# Zileuton, a 5-Lipoxygenase Inhibitor, Attenuates Haemolysate-Induced BV-2 Cell Activation by Suppressing the MyD88/NF-κB Pathway

**DOI:** 10.3390/ijms23094910

**Published:** 2022-04-28

**Authors:** Hui-Yuan Su, Yi-Cheng Tsai, Hung-Pei Tsai, Chih-Lung Lin

**Affiliations:** 1Department of Surgery, Division of Neurosurgery, Kaohsiung Medical University Hospital, Kaohsiung 807, Taiwan; latimeria8@hotmail.com (H.-Y.S.); carbugino@gmail.com (H.-P.T.); 2Graduate Institute of Medicine, College of Medicine, Kaohsiung Medical University, Kaohsiung 807, Taiwan; iiidns11@hotmail.com; 3Graduate Institute of Clinical Medicine, College of Medicine, Kaohsiung Medical University, Kaohsiung 807, Taiwan

**Keywords:** 5-lipoxygenase, BV-2 cells, microglia, subarachnoid haemorrhage, zileuton

## Abstract

M1 microglia induce neuroinflammation-related neuronal death in animal models of spontaneous subarachnoid haemorrhage. Zileuton is a 5-lipoxygenase inhibitor that reduces the levels of downstream pro-inflammatory cytokines. This study aimed to investigate whether zileuton inhibits microglial activation and describe its underlying mechanisms. BV-2 cells were exposed to 1 mg/mL haemolysate for 30 min, followed by treatment with different concentrations (5, 10, 15, or 20 μM) of zileuton for 24 h. The cells were then assessed for viability, polarisation, and protein expression levels. Haemolysate increases the viability of BV-2 cells and induces M1 polarisation. Subsequent exposure to high concentrations of zileuton decreased the viability of BV-2 cells, shifted the polarisation to the M2 phenotype, suppressed the expression of 5-lipoxygenase, decreased tumour necrosis factor α levels, and increased interleukin-10 levels. Furthermore, high concentrations of zileuton suppressed the expression of myeloid differentiation primary response protein 88 and reduced the phosphorylated-nuclear factor-kappa B (NF-kB)/NF-kB ratio. Therefore, phenotype reversal from M1 to M2 is a possible mechanism by which zileuton attenuates haemolysate-induced neuroinflammation after spontaneous subarachnoid haemorrhage.

## 1. Introduction

Spontaneous subarachnoid haemorrhage (SAH) is a type of intracranial haemorrhage mostly caused by a ruptured aneurysm. It has a substantially high mortality rate, with half of the patients dying within 30 days of the onset of the disease, and patients who survive suffer from severe and permanent disabilities [1]. The medical cost for patients with spontaneous SAH is more than twice that for patients with ischaemic stroke. Although vasospasm due to SAH was previously considered the most important cause of severe brain injury, an increasing number of studies have demonstrated that prevention of vasospasm following SAH does not significantly improve the prognostic outcome of patients. Therefore, recent studies have focused on the treatment of early brain injury (EBI) immediately after spontaneous SAH [2]. EBI occurs within 72 h of aneurysm rupture, resulting in neuronal death via numerous mechanisms, including the detrimental reaction caused by the blood clots formed following SAH and transient ischaemic attack induced by SAH. Given that EBI usually lasts for several days following spontaneous SAH, the development of approaches to protect neurones from damage has gradually gained attention [3,4].

Microglia are innate immune cells of the central nervous system that generate an immediate immune response after acute brain injury. Microglia are activated and polarised towards either the M1 or M2 phenotype following SAH. The representative cell surface markers for the M1 phenotype include CD14, CD16, CD32, CD40, CD68, and CD86, whereas the representative cell surface markers for the M2 phenotype include CD206 and CD163. Representative inflammatory mediators produced in M1 microglia include interleukin (IL)-1β, IL-6, IL-8, and tumour necrosis factor (TNF)-α. The representative anti-inflammatory mediators produced in M2 microglia include IL-10, IL-4, IL-13, and transforming growth factor (TGF)-β [5,6,7,8,9].

The lipoxygenase family member 5-lipoxygenase (5-LOX) catalyzes the conversion of arachidonic acid to leukotriene, a potent mediator of many inflammatory processes [10]. Leukotriene B4 (LTB4) is a strong chemoattractant that recruits neutrophils to inflammatory sites by binding to the high-affinity LTB4 receptor 1 (BLT1). In vivo, it was shown that nuclear factor-kappa B (NF-κB)-dependent BLT1 mediates LTB4 signalling after SAH. Therefore, BLT1 may participate in the NF-κB-mediated inflammatory process after SAH [11]. A previous study on an animal model of spontaneous SAH showed that the expression of leukotriene LTB4 receptor 1 (BLT1) in microglia was upregulated following haemorrhage, and the upregulated expression of BLT1 was positively correlated with the activation of the NF-κB pathway [11]. In BV-2 microglia induced by oxygen-glucose deprivation and reoxygenation, suppression of the 5-LOX/LTB4 pathway attenuates nuclear translocation of NF-κB and inflammatory damage [12].

The expression of Toll-like receptor 4 (TLR4) in peripheral blood mononuclear cells is upregulated under oxidative stress. TLR4 is a pattern recognition receptor mainly expressed on the cell surface of peripheral macrophages and microglia in the central nervous system. TLR4 can bind to various molecules, including lipopolysaccharides (LPSs), heat shock proteins 60/70, fibrinogen, and oxidised lipids to form complexes that further activate myeloid differentiation primary response protein 88 (MyD88) and toll receptor-associated activator of interferon (TRIF). The resulting activation of IκB kinase (IKK) phosphorylates the inhibitor of NF-κB (IκB), which is subsequently degraded by the proteasome. This process promotes the nuclear translocation of NF-κB, leading to the production of pro-inflammatory mediators such as TNF-α, IL-6, and IL-12. In this process, microglia are activated and polarised towards the M1 phenotype with an enhanced release of pro-inflammatory mediators, further promoting neuronal death [13,14,15]. MyD88 is a well-established inflammatory adaptor protein and an essential downstream protein of the TLR4-signalling pathway. TLR4 binds to damage-associated molecules, including haeme, fibrinogen, and thrombin. After activation of the TLR-4/MyD88/NF-κB pathway, the nuclear translocation of phosphorylated NF-κB induces the production of pro-inflammatory cytokines [16]. The reversal of M1 microglial polarisation is crucial for the treatment of SAH-induced inflammatory responses. The treatment of LPS-exposed rat microglia with BW-B70C, a 5-LOX inhibitor, has been reported previously. Exposure of rat microglia to LPS induced the activation and nuclear translocation of NF-κB. Treatment with BW-B70C prevented microglial activation by inhibiting the nuclear translocation of NF-κB [17].

Currently, zileuton is primarily administered for the clinical treatment of asthma. It is a 5-LOX inhibitor that suppresses the downstream production of 5-LOX products, such as LTB4, LTC4, LTD4, and LTE4. LTB4 recruits neutrophils and eosinophils, whereas LTC4, LTD4, and LTE4 cause bronchial smooth muscle contraction, mucosal secretion, mucosal oedema, enhanced bronchial hyperresponsiveness, and endothelial permeability. Zileuton inhibits the enzymatic activity of 5-LOX and triggers the above-mentioned reactions by reducing the downstream production of 5-LOX products (e.g., LTB4, LTC4, LTD4, and LTE4), thereby achieving a therapeutic effect against asthma [18]. Zileuton is not only a selective 5-LOX inhibitor but has also been shown to suppress the expression of 5-LOX in an animal model of cerebral ischaemia [19]. In vivo, zileuton improves brain oedema, blood–brain barrier disruption, and neurologic function by attenuating neuronal apoptosis through the activation of the phosphatidylinositol 3 kinase (PI3K)/protein kinase B (AKT) signalling pathway after SAH [10].

The present study aimed to confirm that haemolysate exposure leads to the upregulation of 5-LOX expression, activation of the MyD88/NF-κB pathway, and polarisation of BV2 cells towards the M1 phenotype using cell-based experiments. Inhibition of BV-2 cell activation by downregulating the expression of 5-LOX via zileuton was also attempted. We wanted to confirm whether inhibition of 5-LOX and attenuation of LTB4 would inhibit the NF-κB pathway and reverse microglial polarisation from M1 to M2 after exposure to haemolysate. We also evaluated the correlation between the reversal of microglial polarisation and the MyD88/NF-κB pathway.

## 2. Results

### 2.1. Viability of BV2 Cells under Different Concentrations of Zileuton

We determined the survival rate of BV-2 cells treated with different concentrations of zileuton to gain insight into its effects on BV-2 cells. The cell counts showed that there was no significant difference in the survival rate between BV-2 cells treated with zileuton (5, 10, 15, and 20 μM) and BV-2-cells in the control group at 24 h after treatment (Figure 1a).

### 2.2. Viability of Haemolysate-Exposed BV-2 Cells Treated with Different Concentrations of ZILEUTON

We also measured the viability of BV-2 cells treated with different concentrations of zileuton to explore its effects on haemolysate-exposed BV-2 cells. We found that BV-2 cells treated with haemolysate plus a vehicle or with 5, 10, and 15 μM zileuton had significantly higher viability than those in the control group at 24 h after treatment. Haemolysate-exposed BV-2 cells treated with 20 μM zileuton showed significantly lower viability than those treated with the vehicle at 24 h after treatment (*p* < 0.05). In addition, there was no significant difference in the viability of BV-2 cells in the haemolysate + 20 μM zileuton group and those in the control group. These results indicated that exposure to haemolysate induced the proliferation of BV-2 cells, which was suppressed following treatment with 20 μM zileuton. Cells treated with haemolysate plus 20 μM zileuton showed no statistically significant difference in viability relative to the control group (Figure 1b).

### 2.3. Immunofluorescence Staining of CD68 and CD206 Expressed on BV-2 Cells

Herein, we performed immunofluorescence (IF) staining to determine changes in the expression of CD68 and CD206 in BV-2 cells treated with different concentrations of zileuton to determine the polarisation of BV-2 cells. CD68 labelled with red fluorescence was the representative cell surface marker expressed on M1 microglia, whereas CD206 labelled with green fluorescence was the representative cell surface marker expressed on M2 microglia. The IF results revealed that treatment with 10, 15, and 20 μM zileuton did not significantly alter the expression of the cell surface markers CD206 and CD68 in BV-2 cells compared with the control group (Figure 2a). To determine the polarisation of haemolysate-exposed BV-2 cells following treatment with different concentrations of zileuton, changes in the expression of CD206 and CD68 in haemolysate-exposed BV-2 cells were also observed by IF staining. The IF results showed that the expression level of CD206 on the cell surface of haemolysate-exposed BV-2 cells treated with the vehicle as well as 10, 15, and 20 μM zileuton was higher than that in the control group. However, the expression level of CD68 on the surface of haemolysate-exposed BV-2 cells treated with the vehicle and 10 μM zileuton was higher than that in the control group. The IF results also showed that the haemolysate + 20 μM zileuton group had a higher cell surface expression level of CD206 and a lower cell surface expression level of CD68 than the haemolysate + vehicle group (Figure 2b,c). To reveal the polarisation trend of BV-2 cells, the ratio of the number of CD68-labelled cells to that of CD206-labelled cells was calculated to determine the M1/M2 ratio of BV-2 cells. The haemolysate + 20 μM zileuton group displayed a lower CD68+/CD206+ ratio than the haemolysate + vehicle group. The comparison of the M1/M2 ratio between haemolysate-exposed BV-2 cells treated with different concentrations (10, 15, and 20 μM) of zileuton and those treated with the vehicle showed that the haemolysate + 20 μM zileuton group had a lower M1/M2 ratio of BV-2 cells than the haemolysate + vehicle group (Figure 2d).

### 2.4. Western Blot Results for IL-10, TNF-α, 5-LOX, MyD88, and NF-κB

IL-10 is a representative anti-inflammatory cytokine expressed in activated BV-2 cells with an M2 phenotype. Western blot results revealed that the haemolysate + vehicle, haemolysate + 5 μM zileuton, and haemolysate + 10 μM zileuton groups did not differ significantly from the control group in terms of IL-10 expression. However, the haemolysate + 15 μM zileuton and haemolysate + 20 μM zileuton groups showed increased expression of IL-10 compared to the control and haemolysate + vehicle groups (Figure 3a,b). TNF-α is a representative proinflammatory cytokine expressed in activated BV-2 cells with the M1 phenotype. The Western blot results revealed that haemolysate-exposed BV-2 cells treated with the vehicle as well as 5, 10, and 15 μM zileuton had a higher expression level of TNF-α than BV-2 cells in the control group. The haemolysate + 20 μM zileuton group showed a reduced expression level of TNF-α compared to the haemolysate + vehicle group. However, there was no significant difference in TNF-α expression levels between the haemolysate + 20 μM zileuton group and the control group (Figure 3a,c).

The expression level of 5-LOX in haemolysate-exposed BV-2 cells treated with the vehicle and 5 μM zileuton was higher than that in BV-2 cells in the control group 24 h after treatment. The haemolysate + 20 μΜ zileuton group displayed a lower expression level of 5-LOX than the haemolysate + vehicle group. In addition, there was no significant difference in the expression level of 5-LOX between the haemolysate + 20 μΜ zileuton group and the control group (Figure 4a,b).

The expression of MyD88 in BV-2 cells in the haemolysate + vehicle group was significantly higher than that in the control group. After treatment with 5, 10, and 15 μM zileuton, the expression level of MyD88 did not decrease significantly compared with that in the haemolysate + vehicle group. However, the haemolysate + 20 μM zileuton group exhibited a reduced expression level of MyD88 compared to the haemolysate + vehicle group. In addition, there was no significant difference in the expression level of MyD88 between the haemolysate + 20 μM zileuton and control groups (Figure 4a,c). Changes in the proportion of activated NF-κB in different groups were determined based on the ratio of phospho-NF-κB (pNF-κB) to total NF-κB (pNF-κB/NF-κB ratio), measured via Western blotting. The haemolysate + vehicle group showed a significantly higher pNF-κB/NF-κB ratio than that of the control group. After treatment with different concentrations of zileuton, the pNF-κB/NF-κB ratio in the haemolysate-exposed BV-2 cells treated with 10 and 20 μM zileuton was lower than in those treated with the vehicle, but there was no significant difference compared to the control group (Figure 4a,d).

### 2.5. Enzyme-Linked Immunosorbent Assay (ELISA) Results for LTB4, IL-1β, TNF-α, IL-10, and TGF-β

The concentration of LTB4 in the haemolysate + vehicle group was higher than that of the control group. Following treatment with 5, 10, 15, and 20 μM of zileuton, the haemolysate-exposed BV-2 cells showed a lower concentration of LTB4 than those treated with the vehicle. In addition, there was no significant difference in LTB4 concentration between haemolysate-exposed BV-2 cells treated with zileuton and BV-2 cells in the control group (Figure 5a). The haemolysate-exposed BV-2 cells treated with the vehicle and zileuton (5, 10, 15, and 20 μM) had a higher concentration of IL-1β than BV-2 cells in the control group. The haemolysate-exposed BV-2 cells treated with 10, 15, and 20 μM zileuton exhibited a lower concentration of IL-1β than those treated with the vehicle (Figure 5b). The haemolysate-exposed BV-2 cells treated with the vehicle as well as 5, 10, 15, and 20 μΜ of zileuton had an elevated concentration of TNF-α compared with BV-2 cells in the control group. The haemolysate-exposed BV-2 cells treated with 15 and 20 μM of zileuton showed a lower concentration of TNF-α than those treated with the vehicle (Figure 5c). The haemolysate-exposed BV-2 cells treated with the vehicle as well as 5, 10, 15, and 20 μΜ of zileuton also showed an increased concentration of IL-10 compared to the BV-2 cells in the control group. After treatment with 10, 15, and 20 μΜ of zileuton, the haemolysate-exposed BV-2 cells showed a higher concentration of IL-10 than those treated with the vehicle (Figure 5d). Additionally, haemolysate-exposed BV-2 cells treated with the vehicle as well as 5, 10, 15, and 20 μΜ of zileuton showed an elevated concentration of TGF-β compared with the BV-2 cells in the control group. The concentration of TGF-β in haemolysate-exposed BV-2 cells increased following treatment with 10, 15, and 20 μΜ of zileuton compared to that in cells treated with the vehicle (Figure 5e).

## 3. Discussion

Spontaneous SAH has a high mortality rate. Ruptured aneurysms cause approximately 85% of spontaneous SAH cases are caused by ruptured aneurysms [1]. Haemorrhage due to a ruptured aneurysm contributes to approximately 5–7% of stroke cases. Most patients with ruptured aneurysms are in the prime of their lives at approximately 55 years of age, and are often the backbone of the family and society [20]. The survival rate of patients with spontaneous SAH caused by ruptured aneurysms has increased by approximately 17% in the past few decades owing to the availability of therapeutic alternatives to surgery, such as endovascular embolisation of aneurysms, as well as the improvement in care quality in the intensive care unit. However, only about half of the patients with spontaneous SAH survive, often developing severe neurological sequelae [1]. In addition, only approximately two-thirds of patients who survive are capable of self-care within one year after the onset of spontaneous SAH [21].

Numerous drugs have been employed in previous clinical trials for the treatment of vasospasm after SAH; however, the reduced incidence rate of vasospasm does not seem to improve neurological prognosis [22]. Current therapeutic strategies focus on the prevention of neuronal death, which leads to neurological defects in the brain during EBI following SAH. Advanced neuroimaging techniques, such as phosphorous magnetic resonance spectroscopy, have provided insights into EBI after SAH by detecting changes in energy metabolism in different cerebral territories [23]. Although there is still no consensus on the time course of EBI, after reviewing the literature, we adopted the following definition of EBI: an event that occurs within the first 72 h of SAH [20]. Delayed brain injury represents a series of severe pathological changes that are currently considered sequelae of EBI after SAH. In addition, vasospasm is no longer considered the cause of delayed brain injury but one of its clinical manifestations. Therefore, EBI caused by SAH is believed to be the trigger for delayed cerebral vasospasm and ischaemic nerve injury [24]. It remains unclear whether prevention or attenuation of EBI can reduce the severity of nerve injury by decreasing the incidence of delayed brain injury. There is evidence that supports the theory that EBI intervention prevents delayed neurologic deficits. In vivo, anti-CD47 binds to CD47, an integrin-associated protein expressed on the surface of erythrocytes, to improve neuroprotection by accelerating erythrocyte-phagocytosis [25]. In vivo and in vitro overexpression of S-nitrosoglutathione reductase attenuates nitrosative stress to achieve neuroprotection in EBI after SAH [26].

Previous studies have explored numerous drugs or compounds that inhibit neuronal death during EBI in cellular and animal models, but these have not yielded excellent clinical outcomes in humans [27,28,29]. Several free radicals, which are generated via inflammatory responses induced by subarachnoid blood clots in spontaneous SAH, trigger the lipid peroxidation of polyunsaturated fatty acids in the brain. The resulting lipid peroxides enhance the production of downstream inflammatory mediators leading to neuronal death. Given these responses, a substantial number of pharmacological studies have focused on identifying effective inhibitors of spontaneous SAH-induced lipid peroxidation [30]. Previous studies on cellular models of intracerebral haemorrhage (ICH) have confirmed that hemin induces the upregulation of the expression of 5-LOX and 5-LOX-activating proteins in the nuclear membrane. The 5-LOX-related metabolite, 5-hydroxyeicosatetraenoic acid, is converted to a series of leukotrienes, which are inflammatory mediators that induce cell death. Previous experiments have demonstrated that 5-LOX inhibitors, including zileuton, BW B70, and BW A4C, improve the survival rate of neurones following haemorrhagic stroke in mice [31].

Reversal of M1 polarisation plays a key role in preventing spontaneous SAH-induced nerve injury. For example, activation of retinoic acid receptor (RAR)-α with AM80, a selective agonist of RAR-α, could suppress nuclear translocation of NF-κB and reverse M1 polarisation to the M2 phenotype [32]. Targeting M1 microglia is an effective method to suppress neuroinflammation after SAH. In vivo, L-N6-(1-iminoethyl)-lysine (L-NIL), an inducible nitric oxide synthase (iNOS) inhibitor, inhibits iNOS expression and promotes ferroptosis in M1 microglia to decrease the release of pro-inflammatory molecules such as TNF-α, IL-6, and IL-1β [33]. It has been reported that BW-B70C is a 5-LOX inhibitor that can block the nuclear translocation of NF-κB in LPS-exposed rat microglia, thereby suppressing the production of downstream pro-inflammatory cytokines [17]. Zileuton is a clinical 5-LOX inhibitor widely administered to patients with asthma. The mechanism underlying its therapeutic effects is attributed primarily to its anti-inflammatory activity. Previous studies on animal models of ischaemic stroke demonstrated that zileuton improves the neurological prognosis of rats by inhibiting inflammatory responses via activation of the PI3K/AKT pathway [34]. Zileuton inhibits inflammatory responses and neuronal death by activating the PI3K/AKT pathway in an animal model of spontaneous SAH [10].

Our preliminary experiments revealed that exposure to haemolysate triggered microglial proliferation, which was subsequently suppressed by zileuton. The viability of cells treated with 20 μM zileuton did not differ significantly from that of cells in the control group. Activated microglia have been reported to accumulate and proliferate in the injured brain area [35,36,37,38,39,40]. Further cell-based experiments showed that haemolysate could induce the overexpression of 5-LOX in BV-2 cells, and the expression of 5-LOX was inhibited by zileuton. We found that the exposure of BV-2 cells to haemolysate enhanced the expression of MyD88, which was subsequently suppressed following treatment with 20 μM zileuton. The elevated ratio of activated NF-κB (i.e., pNF-κB) to total NF-κB induces the activation of BV-2 cells with an increased tendency toward M1 polarisation and enhances the production of downstream pro-inflammatory cytokines such as TNF-α [41]. In the present study, exposure to haemolysate increased the proportion of pNF-κB in BV-2 cells, leading to the polarisation of BV-2 cells towards the M1 phenotype. However, 20 μM zileuton reduced the proportion of pNF-κB in BV-2 cells and reversed the M1 polarisation of BV-2 cells towards the M2 phenotype. These results suggest that the inhibitory effect of zileuton against M1 polarisation of BV-2 cells is achieved by suppressing the expression of 5-LOX and reducing the biosynthesis of its downstream products. Previous studies have reported that cysteinyl leukotriene receptor 2 (CYSLTR2) can affect M1/M2 polarisation of microglia through the NF-κB pathway; specifically, activated CYSLTR2 induces M1 polarisation of microglia by activating the downstream NF-κB pathway. In contrast, inhibition of CYSLTR2 results in the polarisation of cells towards the M2 phenotype by suppressing the downstream NF-κB pathway [42]. The decreased levels of 5-LOX products (e.g., cysteinyl leukotriene) reduced the activation of CYSLTR2 and the NF-κB pathway, thereby inhibiting M1 polarisation of BV-2 cells. The M1/M2 polarisation of activated cells following exposure to haemolysate may be associated with the expression of CYSLTR2.

α-Synuclein is a 140-amino-acid protein expressed in presynaptic cells, and its aberrant aggregation structure or overexpression may cause Parkinson’s disease. A previous cell-based experiment on neuroinflammation showed that α-synuclein-induced activation of TNF-α production in BV-2 cells may be related to the TLR4-mediated PI3K/AKT pathway. Inhibition of PI3K or AKT can reduce the production of the downstream inflammatory mediator TNF-α by preventing the nuclear translocation of NF-κB [43]. Our study confirmed that zileuton can reverse the haemolysate-induced M1 polarisation of BV-2 cells by inhibiting the expression of 5-LOX and inhibiting the haemolysate-induced upregulation of MyD88 expression. In addition, zileuton suppressed the production of pro-inflammatory cytokines by inhibiting the nuclear translocation of NF-κB in BV-2 cells. A previous study on neuroinflammation demonstrated that TNF-α production in BV-2 cells may be associated with the TLR4-mediated PI3K/AKT pathway [43]. In the future, the mechanism by which zileuton affects the viability of BV-2 cells can be confirmed by determining whether it prevents the nuclear translocation of NF-κB by inhibiting PI3K or AKT to reduce the downstream production of TNF-α. The potential association of TLR4-mediated PI3K/AKT with the inhibitory effect of zileuton against the M1 polarisation of BV-2 cells can be indirectly confirmed by determining whether the PI3K/AKT activator or TLR4 agonist can reverse the inhibitory effect of zileuton on the M1 polarisation of BV-2 cells.

During inflammatory responses in the central nervous system, microglia and dendritic cells exposed to different cytokines may be polarised towards different phenotypes [44]. It has been reported that LPS is a TLR4 agonist that can induce the upregulation of CYSLTR2 expression during the LPS-induced maturation of dendritic cells [45]. Animal and cell-based experiments on ischaemic stroke have shown that inhibition of the CYSLTR2-ERK1/2 pathway can reduce the M1 polarisation of microglia [46]. Haem, a component of haemolysates, is also a damage-associated molecule that activates the downstream MyD88/NF-κB pathway after binding to TLR4 [47]. This study demonstrated that haemolysate activates the MyD88/NF-κB pathway and promotes the polarisation of cells towards the M1 phenotype. However, it remains to be confirmed whether haemolysates can upregulate the expression of CYSLTR2 by interacting with TLR4. Previous studies have revealed an association between activation of the CYSLTR2-ERK1/2 pathway and M1 polarisation of cells. Therefore, zileuton may affect the activation of the CYSLTR2-ERK1/2 pathway by suppressing the expression of 5-LOX and decreasing the level of its product (i.e., cysteinyl leukotriene) with a reduced effect on CYSLTR2, thereby inhibiting the M1 polarisation of microglia [46].

A previous cell-based experiment showed that the product of 5-LOX is essential for nuclear translocation, that is, the activation of NF-κB in LPS-exposed macrophages. Animal experiments also revealed that macrophages in LPS-exposed 5-LOX^–/–^ mice had lower nuclear translocation of NF-κB than those in wild-type mice. Thus, 5-LOX plays an important role in the MyD88-dependent activation of NF-κB [48]. A previous study in an animal model of spontaneous SAH showed that the expression of BLT1 in microglia was upregulated following haemorrhage, and the upregulated expression of BLT1 was positively correlated with activation of the NF-κB pathway [11]. In a cell-based experiment, LTB4 induced activation of murine BV-2 cells in an autocrine manner following thrombin exposure. U-75302 is a BLT1 antagonist that inhibits the activation of BV-2 cells and reduces the production of downstream pro-inflammatory cytokines [49]. We inferred that the products of 5-LOX (e.g., LTB4) bind to BLT1 and enhance the production of pro-inflammatory cytokines by promoting the activation of BV-2 cells via MyD88-dependent NF-κB activation. In the present study, we suggest that zileuton (5-LOX inhibitor) prevents the activation of BV-2 cells by inhibiting the expression of 5-LOX in BV-2 cells, which reduces the binding of 5-LOX products binding to the BLT1 receptor (e.g., LTB4).

Our study suggested that high concentrations of zileuton inhibited haemolysate-induced activation and proliferation of BV-2 cells, thereby reducing the production of downstream pro-inflammatory cytokines. It is recognised that zileuton regulates the activation of BV-2 cells through the MyD88/NF-κB pathway, but the specific mechanism remains unclear. In addition, whether upstream regulation is achieved via TLR4, CYSLTR2, or BLT1 remains to be clarified. Co-cultivation of microglia with other neurones or neuroglia will be carried out in future studies to determine whether zileuton exerts the same protective effect on neurones. In vivo, zileuton has been shown to improve neurological function in many central nervous system diseases, including spontaneous SAH, intracerebral haemorrhage, Alzheimer’s disease, ischaemic brain damage, and depression [10,34,50,51,52]. Although there is no clinical indication for its use in central nervous system diseases, zileuton is a Food and Drug Administration-approved therapy for asthma [53]. In the future, zileuton might be valuable for treating spontaneous SAH, a fatal disease currently lacking effective therapy to attenuate severe neurologic deficits.

## 4. Materials and Methods

### 4.1. Cellular Model

BV-2 cells are widely employed in cell-based experiments to explore the viability of microglia because they exhibit many microglial characteristics [54,55]. The BV-2 cells (EP-CL-0493; Elabsciences Biotechnology; Houston, TX, USA) were used in this study and cultured in Dulbecco’s modified Eagle’s medium (DMEM; 11995-056, Gibco) containing 10% foetal bovine serum (FBS; TMS-013-BKR, Merck) and 1.5 μg/L penicillin/streptomycin/neomycin (03-033-1B, Biological Industries) at an atmospheric temperature of 37 °C with high humidity in an incubator supplemented with 5% CO_2_. After inoculation into a medium containing 1 mg/mL of haemolysate for 30 min, the BV-2 cells were treated with different concentrations (5, 10, 15, or 20 μM) of zileuton and harvested 24 h after treatment.

### 4.2. Preparation of Haemolysate

Arterial blood samples were collected from the rats and centrifuged at 800× *g* for 5 min. The resulting supernatant was discarded, and the pellet (erythrocytes) was rinsed three times with phosphate-buffered saline (PBS) (PBS:erythrocyte > 5:1) and lysed using four freeze/thaw cycles. Subsequently, the erythrocyte lysate was diluted 1:1 with PBS and centrifuged at 6000× *g* for 30 min to harvest the supernatant, which contained the haemolysate. The concentration of the haemolysate was determined using Drabkin’s reagent (D5941, Sigma Aldrich).

### 4.3. Treatment with Zileuton

The stock solution of zileuton (HY-14164, MedChemExpress) was used in this study and was prepared in 0.2% dimethyl sulfoxide (DMSO; D8418, Sigma Aldrich) at a concentration of 10 mM. Before the experiment, the stock solution of zileuton was diluted with DMEM containing 10% FBS to 5, 10, 15, or 20 μM. The vehicle (control treatment) contained 0.2% DMSO. In the cell-based experiment, BV-2 cells were exposed to 1 mg/mL haemolysate in the culture medium for 30 min and treated with zileuton for 24 h.

### 4.4. Cell Viability Assay

The Cell Counting Kit-8 (CCK-8; C0005, TargetMol) was used to evaluate the viability of cells treated with haemolysate and different concentrations of zileuton. The aim of the assay was to confirm that haemolysate could induce BV-2 cell proliferation, whereas zileuton attenuated this effect. After being suspended in a culture medium (0.5 mL), the BV-2 cells were inoculated into a 24-well plate at a cell density of 5 × 10^4^ cells/mL and incubated for 24 h before treatment with zileuton. The viability of cells treated with zileuton was assessed using the CCK-8 assay based on the degree of CCK-8 reduction. Briefly, BV-2 cells were inoculated into a culture medium containing 0.5 mg/mL CCK-8 and incubated at 37 °C for 1 h. WST-8 in CCK-8 was reduced by dehydrogenases in BV-2 cells to generate orange-yellow formazan, a metabolite that can be dissolved in the culture medium. The concentration of soluble formazan was determined by measuring the absorbance at 450 nm using a microplate reader (MULTISKAN FC, Thermo Fisher Scientific, Tokyo, Japan). The concentration of formazan was expressed as a percentage of the absorbance of the control.

### 4.5. Immunostaining of Cells

2.5 × 10^4^ cells BV-2 cells were inoculated into an 8-well chamber slide at 500 μL/well and exposed to 1 mg/mL of haemolysate for 30 min before treatment with zileuton at 5, 10, 15, or 20 μM. After 24 h of treatment, the cell culture supernatant was discarded, and cells on the cell culture slide were fixed with 4% paraformaldehyde at room temperature for 30 min, rinsed twice with Tris-buffered saline (TBS), and incubated at 4 °C for 16 h with rabbit anti-CD206 (Proteintech, 1:200) and mouse anti-CD68 (Bio-Rad Laboratories, 1:200) primary antibodies. Subsequently, the cells on the glass slide were rinsed twice with TBS and incubated at room temperature for 90 min with the following secondary antibodies: goat anti-rabbit Alexa Fluor^®^488 (Jackson ImmunoResearch, 1:200) and goat anti-mouse Alexa Fluor^®^594 (Jackson ImmunoResearch, 1:200). After rinsing twice with TBS, the glass slide was mounted using Fluoroshield^TM^ with 4′,6-diamidino-2-phenylindole (DAPI). The stained cells were observed under a fluorescence microscope (BX43, Olympus). The fluorescence intensities of DAPI, CD206, and CD68 were analysed using the ImageJ software (National Institutes of Health, NIH). DAPI was used as the nuclear counterstain.

### 4.6. Western Blotting

1 × 10^5^ BV-2 cells suspended in a 2 mL culture medium were incubated into a 6-well plate for 24 h and cultured with zileuton for 24 h. These cells were harvested and resuspended in ice-cold lysis buffer (M-PER Mammalian Protein Extraction Reagent, 78501, Thermo Fisher Scientific) including protease inhibitors (cOmplete™, Mini, EDTA-free Protease Inhibitor Cocktail, 4693159001, Roche, Basel, Switzerland) and phosphatase inhibitors (PhosSTOP, 4906845001, Roche). The protein concentration in the supernatant was measured using Bradford assay (Protein Assay Dye Reagent Concentrate, 5000006, Bio-Rad Laboratories). An equal amount of cell lysate was separated on a 10% sodium dodecyl sulfate-polyacrylamide gel and electrotransferred onto a polyvinylidene difluoride (PVDF) membrane. The PVDF membrane was then immersed in TBS with 0.1% Tween^®^20 detergent (TBST) and 5% skimmed milk powder at room temperature for 1 h to reduce non-specific antibody binding. Subsequently, PVDF membranes were rinsed with TBST and incubated for 2 h at room temperature with the following primary antibodies:Microglial markers: M1 marker, CD68 (1:500, MCA341R, Serotec); M2 marker, CD206 (1:1000, 18704-1-AP, proteintech).5-LOX (1:1000, ab169755, Abcam), TLR4 (1:1000, PA5-23124, Thermo Fisher Scientific), MyD88 (1:1000, 23230-1-AP, Proteintech), NF-κB (1:1000, 10754-1-AP, Proteintech), pNF-κB (1:1000, 3033, Cell Signalling Technology), IL-10, TNF-α, and β-actin (1:10,000, A5441, Sigma-Aldrich).

After incubation with the primary antibodies, the PVDF membrane was incubated with horseradish peroxidase-conjugated secondary antibody [1:2000, anti-rabbit (111-035-144, Jackson ImmunoResearch) or anti-mouse (AP124P, Chemicon)]. Finally, the PVDF membrane was rinsed and incubated in an enhanced chemiluminescence reagent (PerkinElmer), according to the manufacturer’s instructions. The resulting signal intensity was analysed using ImageJ software (NIH).

### 4.7. ELISA

BV-2 cells were inoculated into a 24-well plate and exposed to 1 mg/mL haemolysate for 30 min, followed by treatment with 5, 10, 15, or 20 μM zileuton for 24 h. The cell culture supernatant was harvested to measure cytokine concentrations using the following ELISA kits: IL-1β (KE10003, Proteintech), TNF-α (KE10002, Proteintech), IL-10 (KE10008, Proteintech), TGF-β (RK00057, ABclonal), and LTB4 (E-EL-0061, Elabscience).

### 4.8. Statistical Analyses

All data are expressed as mean ± standard error of the mean. Differences between groups were analysed using a one-way analysis of variance, followed by Tukey’s or Bonferroni post hoc tests. Statistical analyses were conducted using SPSS Statistics (version 24.0; IBM Inc.). Statistical significance was set at *p* < 0.05.

## 5. Conclusions

Treatment with 20 μM zileuton suppressed haemolysate-induced proliferation and activation of BV-2 cells. Zileuton potentially inhibits haemolysate-induced high expression of 5-LOX and reduces M1 polarisation of BV-2 cells by inhibiting the MyD88/NF-κB pathway, thereby preventing the activation of BV-2 cells. The 5-LOX inhibitor, zileuton, might attenuate haemolysate-induced neuroinflammation after spontaneous SAH.

## Figures and Tables

**Figure 1 ijms-23-04910-f001:**
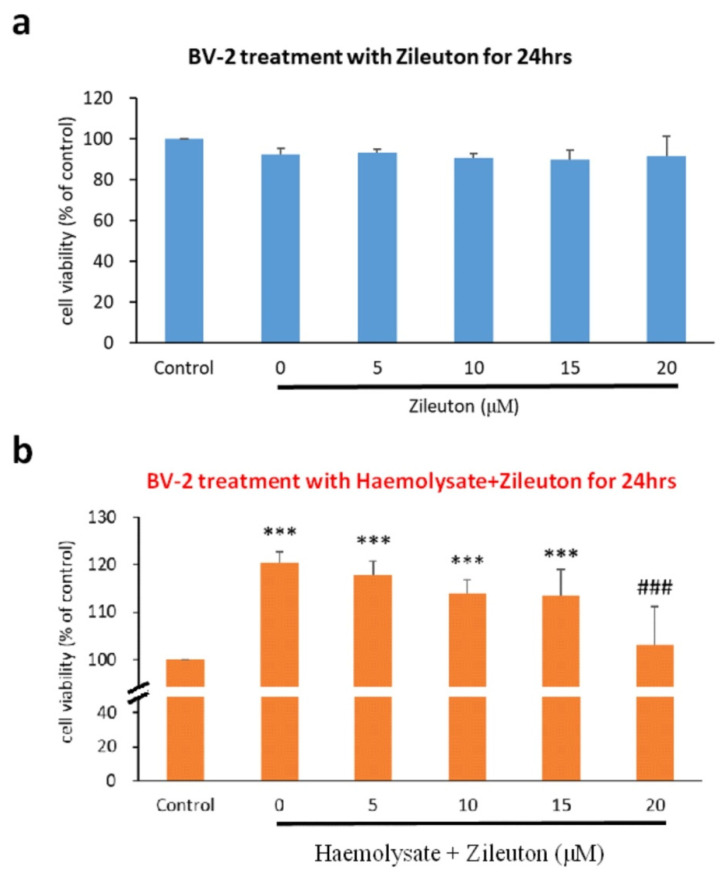
The viability of BV-2 cells. (**a**) The viability of BV-2 cells exposed to zileuton. Differences in the viability of BV-2 cells exposed to different concentrations of zileuton were evaluated via the cell count kit-8 (CCK-8) assay. The *x*-axis represents the control group and vehicle, as well as different concentrations (5, 10, 15, and 20 μΜ) of zileuton; the *y*-axis represents the cell viability expressed as a percentage relative to the control group. Using the different concentrations of zileuton for 24 h did not inhibit the viability of BV-2 cells (*n* = 5). (**b**) The viability of haemolysate-exposed BV-2 cells treated with different concentrations of zileuton. Differences in the viability of haemolysate-exposed BV-2 cells treated with different concentrations of zileuton were evaluated via the CCK-8 assay. The *x*-axis represents the control group and vehicle, as well as different concentrations (5, 10, 15, and 20 μΜ) of zileuton; the *y*-axis represents the cell viability expressed as a percentage relative to the control group. The haemolysate-exposed BV-2 cells treated with the vehicle or 5, 10, and 15 μM of zileuton showed significantly higher viability than BV-2 cells in the control group (*p* < 0.001). The viability of BV-2 cells in the haemolysate + 20 μM zileuton group was significantly lower than that of cells in the haemolysate group (*p* < 0.001). There was no significant difference in the viability of BV-2 cells between the haemolysate + 20 μM zileuton group and the control group. The experimental results revealed that 20 μM of zileuton can inhibit the haemolysate-induced over-proliferation of BV-2 cells (*** *p* < 0.001 compared with control; ^###^
*p* < 0.001 compared with haemolysate group, *n* = 5).

**Figure 2 ijms-23-04910-f002:**
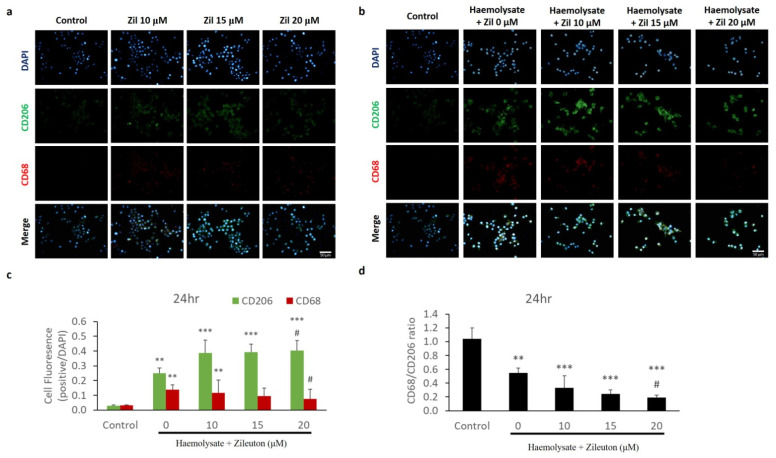
The expression of cell surface markers CD206 (for M2 phenotype) and CD68 (for M1 phenotype) on BV-2 cells as revealed by immunofluorescence (IF) staining. (**a**) Effects of the different concentrations of zileuton on the expression of CD206 and CD68 on haemolysate-free BV-2 cells via IF staining. The *x*-axis represents the control group and different concentrations of zileuton; the *y*-axis represents the IF results for CD206 and CD68. The treatment with zileuton alone did not affect the expression of CD206 and CD68. (**b**) Effects of the different concentrations of zileuton on the expression of CD206 and CD68 on haemolysate-exposed BV-2 cells via IF staining. The *x*-axis represents the control group and different concentrations of zileuton; the *y*-axis represents the IF results for CD206 and CD68. The expressions of CD206 and CD68 on BV-2 cells exposed to haemolysate were upregulated compared to BV-2 cells in the control group. (**c**) IF determination of the proportions of CD206-expressing and CD68-expressing haemolysate-exposed BV-2 cells treated with different concentrations of zileuton. The *x*-axis represents the control group, as well as haemolysate-exposed BV-2 cells treated with the vehicle, 10, 15, and 20 μM of zileuton; the *y*-axis represents the ratio of CD206- and CD68-labelled cells to control cells. The haemolysate-exposed BV-2 cells treated with the vehicle, as well as 10, 15, and 20 μM of zileuton, comprised a greater number of CD206-labelled cells than BV-2 cells in the control group. The number of CD68-labelled BV-2 cells in the haemolysate + vehicle group and haemolysate + 10 μM zileuton group was higher than that in BV-2 cells in the control group. In addition, the haemolysate + 20 μM zileuton group displayed a higher number of CD206-labelled BV-2 cells and a smaller number of CD68-labelled BV-2 cells than the haemolysate + vehicle group (*p* < 0.05) (** *p* < 0.01, *** *p* < 0.001 compared with control; ^#^
*p* < 0.05 compared with haemolysate + vehicle group, *n* = 3). (**d**) IF determination of the ratio of CD68-expressing BV-2 to CD206-expressing BV-2 cells in the presence of haemolysate and different concentrations of zileuton. The *x*-axis represents haemolysate-exposed cells treated with the vehicle, as well as 10, 15, and 20 μM of zileuton; the *y*-axis represents the ratio of CD68-labelled to CD206-labelled cells, which decreased significantly in the haemolysate + 20 μM zileuton group compared with the haemolysate + vehicle group (*p* < 0.05). Overall, the haemolysate + 20 μM zileuton group had a lower proportion of M1-polarised BV-2 cells than the haemolysate + vehicle group (^#^
*p* < 0.05 compared with the haemolysate + vehicle group, *n* = 3).

**Figure 3 ijms-23-04910-f003:**
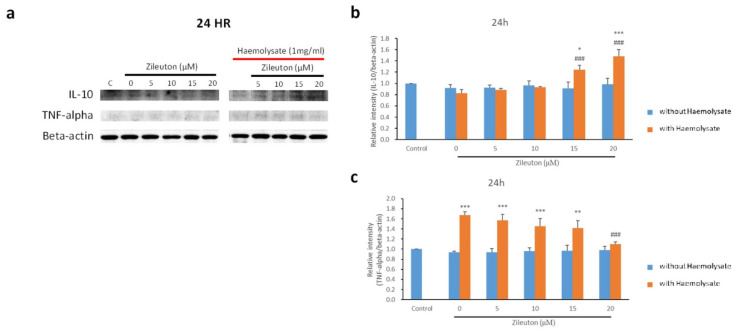
Western blot results for cytokines from BV-2 cells. (**a**) Western blot results for IL-10 and TNF-α. (**b**) Histogram illustrating the Western blot results for IL-10. The differential expression of IL-10 in M1- and M2-polarised BV-2 cells was assessed via Western blot assay to gain insight into the polarisation of BV-2 cells. IL-10 is one of the cytokines expressed on M2 microglia. The *x*-axis represents the control group, and haemolysate-exposed cells treated with the vehicle and different concentrations of zileuton; the *y*-axis represents expression levels as the relative intensity of protein bands that appeared in the Western blot assay. Haemolysate-exposed BV-2 cells treated with 15 and 20 μM of zileuton had a significantly higher level of IL-10 than BV-2 cells in the control and haemolysate + vehicle groups (* *p* < 0.05, *** *p* < 0.001 compared with control; ^###^
*p* < 0.001 compared with haemolysate + vehicle group, *n* = 3). (**c**) Histogram illustrating the Western blot results for TNF-α. Western blot assay was employed to measure the expression level of TNF-α, which is a representative cytokine released by M1 microglia. The haemolysate-exposed BV-2 cells treated with the vehicle, as well as 5, 10, and 15 μM zileuton, had a higher expression level of TNF-α than BV-2 cells in the control group. Following treatment with 20 μM zileuton, the expression level of TNF-α decreased significantly compared with that of BV-2 cells in the haemolysate + vehicle group (*p* < 0.001). However, there was no significant difference between the haemolysate + 20 μM zileuton group and the control group (** *p* < 0.01, *** *p* < 0.001 compared with control; ^###^
*p* < 0.001 compared with the haemolysate + vehicle group, *n* = 3).

**Figure 4 ijms-23-04910-f004:**
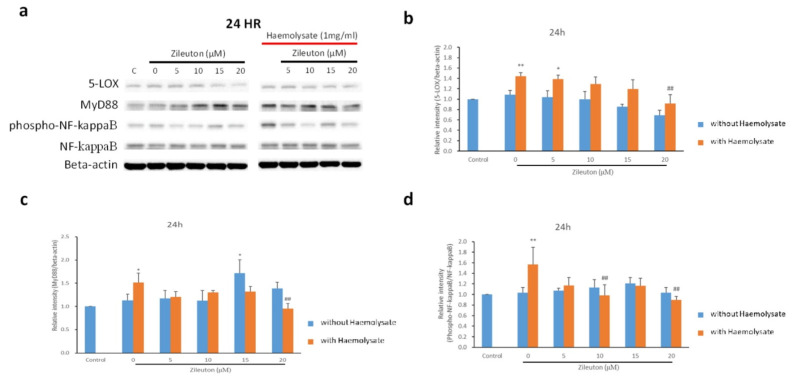
Western blot results for 5-LOX, MyD88, and phosphor-NF-κB (pNF-κB)/NF-κB. (**a**) Western blot results for 5-LOX, MyD88, and pNF-κB/NF-κB. (**b**) Histogram illustrating the Western blot results for 5-LOX expression level. The *x*-axis represents the control group, haemolysate + vehicle, and haemolysate + different concentrations of zileuton; the *y*-axis represents the expression levels as the relative intensity of protein bands that appeared in the Western blot assay. The haemolysate + vehicle group and the haemolysate + 5 μM zileuton group showed a higher expression level of 5-LOX than the control group. The expression of 5-LOX in the haemolysate + 20 μM zileuton group was significantly lower than that in the haemolysate + vehicle group (*p* < 0.01). There was no significant difference between the haemolysate + 20 μM zileuton group and the control group. The haemolysate-induced overexpression of 5-LOX can therefore be inhibited by a high concentration (20 μM) of zileuton (* *p* < 0.05, ** *p* < 0.01 compared with control; ^##^
*p* < 0.01 compared with haemolysate + vehicle group, *n* = 3). (**c**) Histogram illustrating the Western blot results for MyD88. The haemolysate + vehicle group displayed a significantly higher expression level of MyD88 than the control group (*p* > 0.05). The expression level of MyD88 in the haemolysate + 20 μM zileuton group was significantly lower than that in the haemolysate + vehicle group (*p* < 0.01). There was no significant difference between the haemolysate + 20 μM zileuton group and the control group (* *p* < 0.05 compared with control; ^##^
*p* < 0.01 compared with haemolysate + vehicle group, *n* = 3) (**d**) Histogram illustrating the pNF-κB/NF-κB ratio determined via Western blot assay. The activated NF-κB, i.e., pNF-κB, enhances the production of downstream pro-inflammatory cytokines. Herein, we measured the pNF-κB/NF-κB ratio via Western blot assay to determine the activation level of NF-κB, which is directly proportional to the pNF-κB/NF-κB ratio. The haemolysate + vehicle group had a significantly higher pNF-κB/NF-κB ratio than the control group (*p* < 0.01), while the haemolysate-exposed BV-2 cells treated with 10 and 20 μM of zileuton showed a significantly lower pNF-κB/NF-κB ratio than those treated with the vehicle (*p* < 0.01). However, there was no significant difference between the haemolysate + 10 μM and + 20 μM zileuton and the control groups (** *p* < 0.01 compared with control; ^##^
*p* < 0.01 compared with haemolysate + vehicle group, *n* = 3).

**Figure 5 ijms-23-04910-f005:**
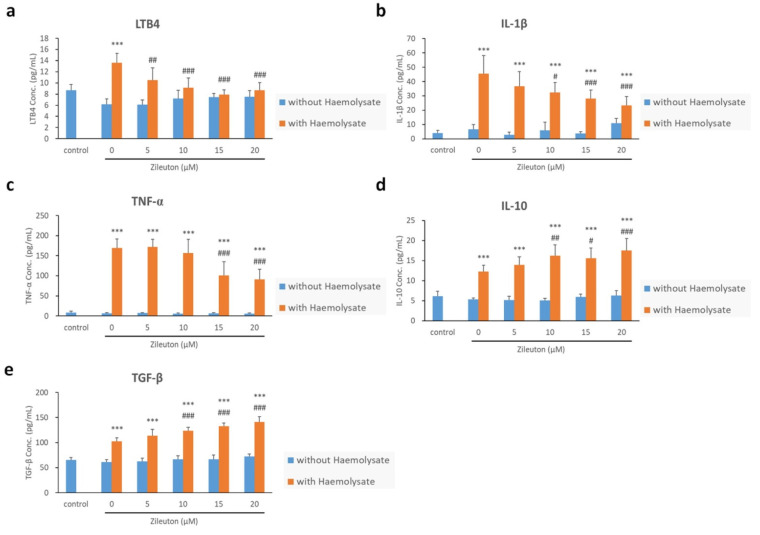
ELISA results for LTB4, IL-1β, TNF-α, IL-10, and TGF-β from the cell culture supernatant. (**a**) ELISA results for LTB4. The concentration of LTB4 in the haemolysate + vehicle group was higher than that in the control group. Following treatment with 5, 10, 15, and 20 μΜ of zileuton, the haemolysate-exposed BV-2 cells showed a lower concentration of LTB4 than those treated with the vehicle. There was no significant difference between haemolysate-exposed, zileuton-treated BV-2 cells, and BV-2 cells in the control group (*** *p* < 0.001 compared with control; ^##^
*p* < 0.01, ^###^
*p* < 0.001 compared with haemolysate + vehicle group, *n* = 3). (**b**) ELISA results for IL-1β. The haemolysate + vehicle group and the haemolysate + zileuton (5, 10, 15, and 20 μΜ) groups had a higher concentration of IL-1β than the control group. The haemolysate-exposed BV-2 cells treated with 10, 15, and 20 μΜ of zileuton displayed a lower concentration of IL-1β than BV-2 cells in the haemolysate + vehicle group (*** *p* < 0.001 compared with control; ^#^
*p* < 0.05, ^###^
*p* < 0.001 compared with haemolysate + vehicle group, *n* = 3). (**c**) ELISA results for TNF-α. The haemolysate-exposed BV-2 cells treated with the vehicle, as well as 5, 10, 15, and 20 μΜ of zileuton, showed elevated concentrations of TNF-α compared with the control group. The haemolysate-exposed BV-2 cells treated with 15 and 20 μΜ of zileuton showed a lower concentration of TNF-α than those treated with the vehicle (*** *p* < 0.001 compared with control; ^###^
*p* < 0.001 compared with haemolysate + vehicle group, *n* = 3). (**d**) ELISA results for IL-10. The haemolysate-exposed BV-2 cells treated with the vehicle, as well as 5, 10, 15, and 20 μΜ of zileuton, had an increased concentration of IL-10 compared with the control group. After being treated with 10, 15, and 20 μΜ of zileuton, haemolysate-exposed BV-2 cells showed a higher concentration of IL-10 than those treated with the vehicle (*** *p* < 0.001 compared with control; ^#^
*p* < 0.05, ^##^
*p* < 0.01, ^###^
*p* < 0.001 compared with haemolysate + vehicle group, *n* = 3). (**e**) ELISA results for TGF-β. The haemolysate-exposed BV-2 cells treated with the vehicle, as well as 5, 10, 15, and 20 μΜ of zileuton, showed an elevated concentration of TGF-β compared with the control group. Additionally, the concentration of TGF-β was higher in the haemolysate-exposed BV-2 cells treated with 10, 15, and 20 of zileuton than in those treated with the vehicle (*** *p* < 0.001 compared with control; ^###^
*p* < 0.001 compared with haemolysate + vehicle group, *n* = 3).

## Data Availability

The datasets used and analysed in the present study are available from the corresponding author upon reasonable request.

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
