# Peer review of "Zileuton, a 5-Lipoxygenase Inhibitor, Attenuates Haemolysate-Induced BV-2 Cell Activation by Suppressing the MyD88/NF-κB Pathway"

_ijms, 2022, doi:10.3390/ijms23094910_

Round 1

Reviewer 1 Report

This study by Su et al. investigates the potential effects of Zileuton on the activation profile of microglia. The authors use BV2 microglia in cell culture and haemolysate to activate the microglia. The inflammatory profile is characterized using immunohistrochemistry, western blot for cell surface markers, and ELISA for cytokine production. There is concern about the statistical analysis. The data presented does not appear to be interpreted correctly in the results. There is also low power in the study, with n=3 being insufficient for in vitro studies like the ones performed here.

Specific comments:

Abstract needs major revisions for English and content. The abstract has too many details regarding experimental design and methods.

For the introduction, authors can look up more references for M1 and M2 microglia markers. For example, CD11b is a surface marker used for both M1 and M2 phenotypes. Also, Arg1 is not a surface marker.

The relevance of 5-LOX is unclear. Need to describe in the introduction what 5-lipoxygenase is and why inhibiting this protein is relevant for microglia activation. Where does 5-LOX fit into the TRL4 activation pathway?

From the introduction it is unclear if Zileuton is an inhibitor of 5-LOX or if it inhibits expression of 5-LOX.

In methods and results, need to state clearly what the vehicle is, since DMSO itself can have anti-inflammatory properties.

It is unclear why cell viability was assessed.

What statistical software was used? In figure 1, it seems unlikely that the Hemolysate+zil 20 group is significantly lower than the vehicle group. The error bars overlap. Related, the statement that there was no statistically significant difference in the viability of BV-2 cells in the haemolysate + 20 μM Zileuton group and those in the control group seems not correct from looking at the graph.

Also, please explain what is meant by higher viability? Is there evidence of proliferation or less cell death?

How was the immunofluorescence images quantified? Please add to methods, and also add microscope used. Figure 2 is hard to interpret. What is the difference between control and 0uM zileuton? The figure legend compares treatments to vehicle, which group is vehicle? The results suggest that Zileuton induces both M1 and M2 microglia, is that correct? Were there any cells that expressed both CD206 and CD68? Please elaborate.

Experimental design comment: n=3 for one replicate is very low. Would prefer if authors did two replicates. Also, please consult a statistician as I believe some of these results should be analyzed in a two-way anova.

The MyD88 and Nfkb western results are too variable to make any conclusions from. Potentially by repeating the experiments a better result can be obtained.

In Figure 5 there does not appear to be a statistically significant effect of Zileuton which makes me question the statistical analysis and interpretations.

Reviewer 2 Report

My suggestion is Major Revision, and my Review Report is as follows: 

Authors present an in-vitro study via cell-based experiments in order to investigate wheather Zileuton, 5-LOX inhibitor, inhibits microglial activation and in that manner prevents early brain injury following subarachnoid hemorrhage. BV-2 cells were exposed to 1 mg/ml of haemolysate for 30
min, followed by treatment with different concentrations of Zileuton. Cell surface markers were visualized using immunofluorescence and protein expression levels by western blot. Immunofluorescence
staining for BV-2 cells at 24 hours post-treatment showed that the haemolysate + 20 μΜ Zileuton

group had a lower ratio of CD68+/CD206+ BV-2 cells than the haemolysate + vehicle group, indicat
ing that the former had a reduced proportion of M1-polarised BV-2 cells.Treatment with 20 μM Zileuton suppressed haemolysate-induced proliferation and M1 polarisation of BV-2 cells.

This manuscripts add to the body of literature on possible positive influence of 5-LOX inhibitors and other drugs, such as statins, on the tissue damage in the subarachnoid haemorrhage. Introduction is well written and Materials and Methods provide satisfactory explanations. Results are conclusive. Materials and methods as well as results should be reviewed by molecular biologist with experience in the field.

However, Discussion is too long and there are several sentences which are repeated from the Introduction. I suggest to divide Discussion into subsections and to shorten it. Potential future applications should be discussed.  There are several important studies on drug treatment of EBI, which were not included into the literature review; I suggest to cite and discuss:

Treichl SA, Ho WM, Steiger R, Grams AE, Rietzler A, Luger M, Gizewski ER, Thomé C, Petr O. Cerebral Energy Status and Altered Metabolism in Early Brain Injury After Aneurysmal Subarachnoid Hemorrhage: A Prospective 31P-MRS Pilot Study. Front Neurol. 2022 Feb 28;13:831537. doi: 10.3389/fneur.2022.831537. PMID: 35295831; PMCID: PMC8919991.

Xu CR, Li JR, Jiang SW, Wan L, Zhang X, Xia L, Hua XM, Li ST, Chen HJ, Fu XJ, Jing CH. CD47 Blockade Accelerates Blood Clearance and Alleviates Early Brain Injury After Experimental Subarachnoid Hemorrhage. Front Immunol. 2022 Feb 25;13:823999. doi: 10.3389/fimmu.2022.823999. PMID: 35281006; PMCID: PMC8915201.

Wang L, Wang Z, You W, Yu Z, Li X, Shen H, Li H, Sun Q, Li W, Chen G. Enhancing S-nitrosoglutathione reductase decreases S-nitrosylation of Drp1 and reduces neuronal apoptosis in experimental subarachnoid hemorrhage both in vivo and in vitro. Brain Res Bull. 2022 Mar 15:S0361-9230(22)00079-X. doi: 10.1016/j.brainresbull.2022.03.010. Epub ahead of print. PMID: 35304287.

Qu W, Cheng Y, Peng W, Wu Y, Rui T, Luo C, Zhang J. Targeting iNOS Alleviates Early Brain Injury After Experimental Subarachnoid Hemorrhage via Promoting Ferroptosis of M1 Microglia and Reducing Neuroinflammation. Mol Neurobiol. 2022 Mar 9. doi: 10.1007/s12035-022-02788-5. Epub ahead of print. PMID: 35262869.

Tian Y, Liu B, Li Y, Zhang Y, Shao J, Wu P, Xu C, Chen G, Shi H. Activation of RARα Receptor Attenuates Neuroinflammation After SAH via Promoting M1-to-M2 Phenotypic Polarization of Microglia and Regulating Mafb/Msr1/PI3K-Akt/NF-κB Pathway. Front Immunol. 2022 Feb 14;13:839796. doi: 10.3389/fimmu.2022.839796. PMID: 35237277; PMCID: PMC8882645.

Round 2

Reviewer 1 Report

In this revised manuscript, the authors have addressed my prior concerns. I appreciate the clarifications for the methods used as well as the additional replicates performed.

Reviewer 2 Report

The authors have sufficiently answered my remarks, I suggest furthermore the technical part of the paper, especially methods and materials to be revised through molecular biologist. 
